# Pregnancy outcomes after snakebite envenomations: A retrospective cohort in the Brazilian Amazonia

**Thaís P. Nascimento**[1], **Alexandre Vilhena Silva-Neto**[2,3], **Djane Clarys Baia-da-Silva**[1,2,3,4], **Patrícia Carvalho da Silva Balieiro**[2,3], **Antônio Alcirley da Silva Baleiro**[1], **Jacqueline Sachett**[2,5], **Lisele Brasileiro**[2,3], **Marco A. Sartim**[2,3], **Flor Ernestina Martinez-Espinosa**[1,3], **Fan Hui Wen**[5], **Manuela B. Pucca**[6], **Charles J. Gerardo**[7], **Vanderson S. Sampaio**[3], **Priscila Ferreira de Aquino**[1], **Wuelton M. Monteiro**[2,3]*

1 Leônidas & Maria Deane Institute, Oswaldo Cruz Foundation, Manaus, Amazonas, Brazil, 2 Amazonas State University, Manaus, Amazonas, Brazil, 3 Dr. Heitor Vieira Dourado Tropical Medicine Foundation, Manaus, Amazonas, Brazil, 4 Amazonas Federal University, Manaus, Amazonas, Brazil, 5 Butantan Institute, São Paulo, Brazil, 6 Roraima Federal University, Boa Vista, Roraima, Brazil, 7 Department of Emergency Medicine, Duke University School of Medicine, Durham, North Carolina, United States of America

* wueltonmm@gmail.com

**Data Availability Statement:** All relevant data are within the manuscript and its Supporting Information files.

## Abstract

Snakebite envenomations (SBEs) in pregnant women can result in adverse maternal or neonatal effects, such as abortion, placental abruption, preterm labor, fetal malformations, and maternal, fetal or neonatal deaths. Despite the high incidence of SBEs in the Brazilian Amazon, there is no literature on the impact of SBEs on pregnancy outcomes. The objective of this study was to describe clinical epidemiology and outcomes associated with SBEs in women of childbearing age and pregnant women in the state of Amazonas, Western Brazilian Amazon, from 2007 to 2021. Information on the population was obtained from the Reporting Information System (*SINAN*), Mortality Information System (*SIM*) and Live Birth Information System (*SINASC*) for the period from 2007 to 2021. A total of 36,786 SBEs were reported, of which 3,297 (9%) involved women of childbearing age, and 274 (8.3%) involved pregnant women. Severity (7.9% in pregnant versus 8.7% in non-pregnant women) ($P = 0.87$) and case-fatality (0.4% in pregnant versus 0.3% in non-pregnant women) rates were similar between groups ($P = 0.76$). Pregnant women who suffered snakebites were at higher risk for fetal death (OR: 2.17, 95%CI: 1.74–2.67) and neonatal death (OR = 2.79, 95%CI: 2.26–3.40). This study had major limitations related to the completeness of the information on the pregnancy outcomes. Although SBE incidence in pregnant women is low in the Brazilian Amazon, SBEs increased the risk of fetal and neonatal deaths.

## Author summary

Snakebite envenomations (SBEs) are a health problem in tropical and subtropical countries. In endemic areas, this health problem more frequently involves men engaged in agricultural activities. However, in many areas of the world, women of childbearing age

**Funding:** J.S., M.B.P., and W.M.M. were funded by Conselho Nacional de Desenvolvimento Científico e Tecnológico (CNPq productivity scholarships). W. M.M. was funded by Fundação de Amparo à Pesquisa do Estado do Amazonas (PRÓ-ESTADO, call 011/2021 - PCGP/FAPEAM, call 010/2021 - CT&I ÁREAS PRIORITÁRIAS, call 003/2022 - PRODOC/FAPEAM, and POSGRAD) and by the Ministry of Health, Brazil (proposal No. 733781/19-035). The funders had no role in study design, data collection and analysis, decision to publish, or preparation of the manuscript.

**Competing interests:** The authors have declared that no competing interests exist.

are also very affected by snakebites, as they also actively participate in activities that put them at risk of encountering snakes. In this context, SBEs can occur during pregnancy, with deleterious consequences for both the mother and the fetus or neonate. However, the little that is known about the consequences of SBEs in pregnancy comes from case reports. In this study, we linked the surveillance databases Snakebites Reporting Information System (*SINAN*), Mortality Information System (*SIM*) and Live Birth Information System (*SINASC*) of the state of Amazonas, Western Brazilian Amazon, to gain a more integrative view of the problem in the region. Although severity and case-fatality rates of SBEs were similar between pregnant and non-pregnant women, pregnant women who suffered SBEs were at higher risk for fetal and neonatal deaths.

## Introduction

Snakebite envenomation (SBE) is a serious global public health problem that per year affects approximately 2.7 million people and causes more than 100,000 deaths, predominantly in tropical, low and middle-income countries [1]. Over 25,000 snakebites occurred in Brazil in 2021, with, depending on the region, about 70 to 90% of these being caused by lanceheads (*Bothrops* spp.) [2]. In the Amazon region of Brazil, cases have an incidence that is four times higher than in the rest of the country [3,4]. The burden of SBEs has not received proper attention from public health community, development agencies and governments, but is currently properly categorized as a Category A neglected tropical disease [5]. Most SBE reporting systems are fragile and underestimate the actual numbers and case fatalities [6], thus better epidemiological surveillance is necessary to assess the extent of this important public health problem so as to improve prevention and treatment interventions. Most deaths and sequelae from SBEs are preventable by interventions such as early administration of antivenom [7,8].

Generally, SBEs caused by *Bothrops* snakes are characterized by local (pain, swelling, blisters, and bleeding from the bite site) and systemic (abnormal clotting, spontaneous bleeding and kidney failure) manifestations [9]. Other complications include bacterial infections, necrosis, compartment syndrome, and amputations [10–12]. To prevent severe cases, antivenom must be administered quickly. In the Brazilian Amazon, antivenom is available only in urban health units and time to medical care may take hours or even days. In this scenario of poor access to medical care, the use of traditional practices, such as herbal preparations, incisions at the bite site, application of black stones, and tourniquets, are often used [13,14], which delay the effective care and contribute to severity and a greater number of deaths [4,15].

As observed in other parts of the world, SBEs in Brazil mainly affect working males in rural settings [4,16]. However, women of childbearing age are also exposed to SBEs, especially in the rural areas, and SBEs in pregnant women have been reported [3,15]. When pregnant women suffer SBEs, maternal or neonatal pathological effects may be observed, such as abortion, placental abruption, preterm labor, fetal malformations, maternal death, and fetal or neonatal death [17–19]. Although these events are important and maternal death has an impact on the family and community, most of the information in the literature is based on case reports and a few case series [20]. Therefore, the absence of robust evidence may mask the actual burden of SBEs in pregnant women and make clinical management difficult.

The objective of this study was to describe clinical epidemiology and outcomes associated with SBEs in women of childbearing age and pregnant women in the state of Amazonas, Western Brazilian Amazon, in cases from 2007 to 2021.

## Material and methods

### Ethics statement

This study was conducted in accordance with the principles of the Declaration of Helsinki and the guidelines of Good Clinical Practice of the International Harmonization Conference. The study was approved by the Ethics Review Board (ERB) of the *Fundação de Medicina Tropical Dr. Heitor Vieira Dourado* (CAAE: 52805821.4.0000.5016). The ERB gave a waiver for informed consent. After database linkages, the final dataset was anonymized before statistical analysis.

### Study area

The state of Amazonas is located in the Western Brazilian Amazon, and comprises an area of 1,559,167.878 km$^2$ (the biggest state in the country), with 62 municipalities. The estimated population of the state is 4,269,995 inhabitants (2021), with 80% living in urban zones and 20% in rural, riverine, and indigenous areas. Approximately 50% of the population lives in the state capital, Manaus. The state has a reduced coverage in terms of highways and roads, and a primarily fluvial transportation system. The state is densely covered by a rainforest that is comprised of the upland forests (*terra firme* forest), floodplains (*várzeas*), and flooded areas (*igapós*).

In 2021, 1,996 SBEs were reported in the state of Amazonas (~50 cases/100,000 inhabitants), with 90% of cases caused by the lancehead snake (*Bothrops atrox*) [9]. In the 62 municipalities of this state, 78 registered health units provide antivenom treatment free of charge. SBEs are compulsorily recorded in structured forms available on-line as part of the Brazilian Ministry of Health's (MoH) Reporting Information System (*SINAN*).

### Study design

This is a concurrent, cohort study, based on surveillance data from SBE patients treated in health units of the state of Amazonas and reported on the *SINAN*, between January 2007 and December 2021. In this study, exposure was defined as an SBE episode, according to the Brazilian MoH guidelines [21]. The predictor variables age, education (years of schooling), self-reported ethnicity, occurrence zone (rural/urban), association with work activities, time from snakebite to medical assistance (in hours), anatomical site of the bite, type of envenomation, local and systemic manifestations, Lee-White clotting test result, severity classification, and antivenom treatment were also used. The variable of ethnicity in the databases is self-reported, and the investigators recognize that the available choices of ethnicity may not fully or accurately reflect the ethnic identity of the patients. Specifically, the term *pardo* (mixed-race) is both widely used, but also rejected by some.

### Outcomes

The present study was designed to estimate the risk of (i) *severe SBE cases*, life-threatening SBEs with severe bleeding, hypotension/shock and/or acute renal failure, as reported to *SINAN* database [21]; (ii) *maternal SBE-related case-fatalities*, defined as deaths reported as X.29, X.28, R98, and R99 according the ICD10-10th revision, as reported to the Mortality Information System (*SIM*) [22]; (iii) *low birth weight*, defined by WHO as weight at birth of < 2,500 grams (5.5 pounds) [23], as reported to Live Birth Information System (*SINASC*); (iv) *preterm birth*, defined as babies born alive before 37 weeks of pregnancy are completed, as reported to *SINASC*. There are sub-categories of preterm birth, based on gestational age: extremely preterm (less than 28 weeks), very preterm (28 to 32 weeks), moderate to late

preterm (32 to 37 weeks) [24]; v) *fetal deaths*, which refer to the intrauterine death of a fetus at any time during pregnancy [25]; (vi) *neonatal deaths*, death of live newborn before the age of 28 complete days [26]; and (vii) *perinatal mortality*, defined as a death from 22 weeks of gestation until 7 days after birth [26].

## Data processing and record linkage strategy

The database variables were standardized by removing special characters such as punctuation, prepositions, and graphic accents. Dates of birth have been changed to "day-month-year" format. In order to only assess information from women of childbearing age (10–49 years) [27], SBEs in males, women outside of childbearing age, dry bites and cases of bites by non-venomous snakes were excluded from the *SINAN* database. Gestational status was assessed in women of childbearing age using the *SINAN*, *SINASC*, and *SIM* databases. Severity classification was obtained from the *SINAN* database. Case fatality was assessed via the *SIM* database. Pregnancy outcomes were also assessed using the information from the *SINASC* and *SIM* databases. The crossing of the data from the *SINAN*, *SIM*, and *SINASC* databases was performed using Record probabilist linkage with the "R" language, on the Rstudio 3.11.1 platform, "RecordLinkage" library and phonetic method and "Levenshtein". Three linkages were made: (i) *SINAN* X *SIM* (pairing in identification of deaths of women of childbearing age), with blocking performed from the date of birth, "patient's name" (*SINAN*) "deceased's name" (*SIM*), and "mother's name" (*SINAN* and *SIM*); (ii) *SINAN* X *SIM* (pairing to identify fetal or perinatal deaths) with the blocking performed from the surnames that were compared with the variables "patient's name" (*SINAN*), "mother's name" (*SIM*) and "city of residence" (*SINAN* and *SIM*); (iii) *SINAN* X *SINASC* (pairing for identification of complications in live births) with blocking of surnames and comparisons between the variables "patient's name" (*SINAN*), "mother's name" (*SINASC*) and "city of residence" (*SINAN* and *SINASC*). We obtained a final selection of pairs identified as likely to be from the same patients by automatic verification, applying a probability threshold (probability > 0.7) for all linkages. Pairs with score values > 0.3 were manually double-checked to identify other true pairs. A final identification was created for all included participants.

## Data analysis

Descriptive statistics were used for demographic variables. Continuous variables presented as mean and standard deviation. Maternal and perinatal cases evolving to death were individually described. Student's t test was used to compare means, and Chi-square or Fisher's exact test were used to compare proportions, as appropriate. Crude odds ratio (OR) with its respective 95% confidence interval (95%CI) was determined in a univariate analysis. Logistic regression was used for the multivariate analyses and the adjusted OR (AOR) with 95% CI were also estimated. A log binomial multivariate generalized linear regression was performed using an automated forward stepwise estimation. All variables that were associated with dependent variables at a significance level of $P < 0.2$ in the univariate analysis were included in the multivariate analysis. Statistical significance was considered when $P < 0.05$ in the Hosmer–Lemeshow goodness-of-fit test. The statistical analyses were carried out using R software (version 4.1.0).

## Results

### Participants' characteristics

There were 36,786 SBEs reported in the state of Amazonas from 2007 to 2021. A total of 3,297 cases occurred in women of childbearing age, of which 274 (8.3%) were pregnant (Fig 1). The

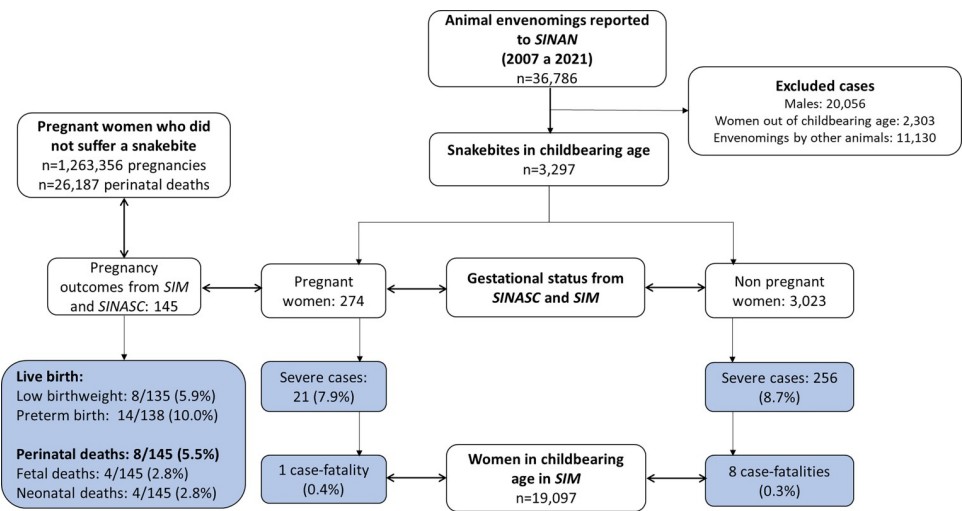

**Fig 1. Study flowcharts, with absolute numbers and frequencies of snakebite envenomations in women of childbearing age and pregnant women, and maternal and perinatal outcomes.** *SINAN*: Case Reporting Information System; *SINASC*: Live Birth Information System; *SIM*: Mortality Information System.

frequency of SBEs during pregnancy was 21.7 SBE cases per 100,000 pregnancies. Mostly sociodemographic and clinical aspects are similar in pregnant and non-pregnant women (Table 1). Mean age of pregnant women was significantly lower than that for non-pregnant women [OR = 0.97 (95%CI 0.96–0.99)]. Most pregnant women affected by SBEs reported ethnicity as either *pardo* (58.9%) or Amerindian (34.1%), had 4–7 years of schooling (62.9%), and lived in rural areas (82.8%). A total of 40% of the SBEs in pregnant women were related to work activities. Regarding time to medical assistance, 51.1% were admitted to health units within 3 hours after the snakebite occurring. Bites were reported mostly in the lower limbs (87.1%). Envenomations caused by the lance head snake (*Bothrops*) were diagnosed in 73.7% of the cases. Pain, edema, and ecchymosis were the major local manifestations. The most common local complications were secondary bacterial infections (2.7%) and necrosis (2.0%). Unclottable blood was reported in 39.8% of the cases. Acute kidney injury was the major systemic complication (2.4%). A total of 47.3% of the cases presented moderate severity, and 88.0% received antivenom treatment.

## Maternal outcomes

The frequency of severe cases was similar between pregnant (7.9%) and non-pregnant (8.7%) women with SBEs [OR = 0.96 (95%CI 0.59–1.55)] (Table 1). Case-fatality was 0.4% (1/274) in pregnant women and 0.3% (8/3,023) in non-pregnant women, with no statistical difference [OR = 1.38 (95%CI 0.17–11.03)] (Table 1).

Fatal SBEs in women of childbearing age are described in Table 2. Out of the nine women, one was in the first trimester of pregnancy. This pregnant woman was 39 years old, had Amerindian ethnicity, and was a resident of the municipality of Tabatinga. She was bitten on the foot by a *Bothrops* snake. On hospital admission, the case was classified as severe and 8 vials of *Bothrops* antivenom were prescribed. She died one day after admission. Unfortunately, clinical description and cause of death were not available in the databases. Eight non-pregnant women also died during the study period; five were of Amerindian ethnicity, and ages ranged from 26 to 49 years. Major local complications were secondary bacterial infections and necrosis. Acute kidney failure was the most common systemic complication during hospitalization. Acute

**Table 1. Demographic and clinical aspects between pregnancy and women of childbearing age.**

| Variables | Total, n, % n = 3,297 | Pregnant, n,% n = 274 | Non-pregnant, n,% n = 3,023 | P | OR (95%CI) |
|---|---|---|---|---|---|
| Mean age (±SD) | 28.3 (10.7) | 25.7 (9.3) | 28.5 (10.8) | <0.01 | 0.97 (0.96–0.99) |
| Ethnicity (n = 3,262) | | | | | |
| White | 110 (3.4) | 2.6 | 3.4 | 1 | 1 |
| Black | 84 (2.6) | 3.7 | 2.5 | 0.18 | 1.99 (0.72–5.46) |
| Asian | 13 (0.4) | 0.7 | 0.4 | 0.25 | 2.67 (0.49–14.50) |
| *Pardo* | 2,205 (67.6) | 58.9 | 68.4 | 0.74 | 1.14 (0.52–2.50) |
| Amerindians | 850 (26.1) | 34.1 | 25.3 | 0.15 | 1.78 (0.80–3.96) |
| Education (in years) (n = 2,357) | | | | | |
| 0–4 | 227 (9.6) | 11.9 | 9.4 | 1 | 1 |
| 4–7 | 1,366 (58.0) | 62.9 | 57.5 | 0.54 | 0.86 (0.55–1.37) |
| 8–15 | 488 (20.7) | 19.3 | 20.8 | 0.25 | 0.73 (0.43–1.25) |
| 16–18 | 251 (10.6) | 5.4 | 11.1 | 0.01 | 0.39 (0.19–0.81) |
| >18 | 25 (1.1) | 0.5 | 1.1 | 0.32 | 0.35 (0.05–2.72) |
| Occurrence zone (n = 3.261) | | | | | |
| Urban | 479 (14.7) | 16.5 | 14.5 | 1 | 1 |
| Rural | 2,738 (84.0) | 82.8 | 84.1 | 0.41 | 0.87 (0.62–1.22) |
| Periurban | 44 (1.3) | 0.7 | 1.4 | 0.29 | 0.46 (0.11–1.96) |
| Work-related snakebite (n = 3,196) | 1,195 (37.0%) | 40.0 | 37.0 | 0.53 | 1.08 (0.83–1.41) |
| Time from bite to medical assistance (in hours) (n = 3,160) | | | | | |
| 0–3 | 1,561 (49.4) | 51.1 | 49.2 | 1 | 1 |
| 3–12 | 1,078 (34.1) | 31.6 | 34.3 | 0.40 | 0.88 (0.66–1.17) |
| 12–24 | 282 (8.9) | 8.6 | 8.9 | 0.75 | 0.93 (0.58–1.47) |
| >24 | 239 (7.6) | 8.6 | 7.5 | 0.64 | 1.11 (0.70–1.77) |
| Bite site (3,271) | | | | | |
| Head | 32 (1.0) | 1.5 | 0.9 | 1 | 1 |
| Upper limbs | 404 (12.4) | 11.0 | 12.5 | 0.31 | 0.56 (0.18–1.71) |
| Lower limbs | 2,827 (86.4) | 87.1 | 86.4 | 0.41 | 0.64 (0.22–1.85) |
| Trunk | 8 (0.2) | 0.4 | 0.2 | 1 | 1 (0.10–10.41) |
| Type of envenomings (n = 3,297) | | | | | |
| *Bothrops* | 2,465 (74.8) | 73.7 | 74.9 | 1 | 1 |
| *Crotalus* | 14 (0.4) | 0.0 | 0.5 | 1 | 1 |
| *Micrurus* | 15 (0.5) | 0.0 | 0.5 | 1 | 1 |
| *Lachesis* | 459 (13.9) | 14.6 | 13.9 | 0.71 | 1.06 (0.75–1.52) |
| Unknown | 344 (10.4) | 4.4 | 3.1 | 0.27 | 1.41 (0.76–2.62) |
| Local manifestations (3,051) | | | | | |
| Pain | 2,985 (97.8) | 98.8 | 97.7 | 0.27 | 1.93 (0.60–6.18) |
| Edema | 2,477 (81.3) | 78.7 | 81.6 | 0.26 | 0.83 (0.61–1.14) |
| Ecchymosis | 497 (16.4) | 16.3 | 16.4 | 0.97 | 0.99 (0.70–1.41) |
| Necrosis | 64 (2.1) | 2.0 | 2.1 | 0.23 | 2.14 (0.66–7.01) |
| Secondary bacterial infection | 178 (5.9) | 2.7 | 6.1 | 0.03 | 0.43 (0.20–0.92) |
| Compartment syndrome | 26 (0.9) | 0.8 | 0.9 | 0.89 | 0.90 (0.21–3.87) |
| Systemic manifestations (3,018) | | | | | |
| Acute kidney injury | 69 (2.3) | | 3.2 | 0.88 | 1.05 (0.45–2.46) |
| Respiratory failure | 11 (0.4) | 0.8 | 0.3 | 0.24 | 2.48 (0.53–11.55) |
| Sepsis | 6 (0.2) | 0.4 | 0.2 | 0.46 | 2.22 (0.25–19.15) |
| Shock | 6 (0.2) | 0.0 | 0.2 | . . . | . . . |

*(Continued)*

**Table 1.** (Continued)

| Variables | Total, n, % n = 3,297 | Pregnant, n,% n = 274 | Non-pregnant, n,% n = 3,023 | P | OR (95%CI) |
|---|---|---|---|---|---|
| Unclottable blood (2,090) | 863 (41.3) | 39.8 | 41.4 | 0.67 | 0.93 (0.67–1.28) |
| Severity classification (3,178) | | | | | |
| Mild | 1,500 (47.2) | 44.7 | 47.4 | 1 | 1 |
| Moderate | 1,401 (44.1) | 47.3 | 43.8 | 0.30 | 1.14 (0.88–1.49) |
| Severe | 277 (8.7) | 7.9 | 8.7 | 0.87 | 0.96 (0.59–1.55) |
| Antivenom treatment (n = 3,219) | 2,847 (88.4) | 88.0 | 88.5 | 0.58 | 1.02 (0.93–1.12) |
| Deaths from snakebites (3,297) | 9 (0.3%) | 1 (0.4%) | 8 (0.3%) | 0.76 | 1.38 (0.17–11.03) |

respiratory failure, sepsis, and acute kidney failure were the main immediate causes of death. Six women received antivenom treatment and two died without medical care (Table 2).

## Perinatal outcomes

Among the 145 pregnant women affected by SBEs who had information on pregnancy outcomes, 24 (16.5%) had perinatal complications during pregnancy. A total of 129 (47.1%) pregnant women with reports of SBEs during pregnancy had no pregnancy outcome information available in the databases. No statistical difference was found between pregnant women with outcome information or pregnant women with missing data on outcome, in relation to age, ethnicity, literacy, occurrence area (rural or urban), and time to medical assistance after the SBE (P>0.01). No sociodemographic or clinical variables were associated with poor maternal outcomes among pregnant women affected by SBEs (Table 3).

Pregnant SBE patients had an incidence of 5.9% low birth weight and 10.0% preterm birth of the live births, and were not associated with an increased risk of low birth weight and preterm birth when compared to non-SBE pregnant patients. Frequency of perinatal death was 5.6%, with 2.8% fetal and 2.8% neonatal deaths (Table 4). Comparing pregnant women affected by SBEs with all the other pregnant women, SBEs during pregnancy was significantly associated with fetal death [P<0.01; AOR = 2.24 (95% CI 1.80–2.76)] and neonatal death [P<0.01; AOR = 2.89 (95% CI 2.34–3.52)], after adjusting by age, schooling, and number of antenatal consultations (Table 4).

Two fetal deaths occurred at 22–27 weeks, one at 32–36 weeks, and one at ≥42 weeks. Three of the cases were attributed to *Bothrops*, and these three received antivenom. All of these SBEs were reported in the first trimester of pregnancy. The four cases were classified as mild in severity. Causes of fetal death are detailed in Table 5. No maternal deaths were reported in this group. Vaginal delivery was reported in the four cases of fetal death.

Neonatal deaths occurred from 1 to 16 hours after birth. Gestational age ranged from <22 weeks (in an extreme low birth weight, 315 grams, newborn) to 37–41 weeks (birth weight 3,200 grams). Three pregnant women were bitten by *Bothrops* and one by *Lachesis* snakes. In terms of clinical severity, one patient was mild, two were moderate, and one was severe. All four women received antivenom treatment. In addition, no maternal deaths were reported in this group. Cesarian delivery was reported in two cases (Table 5).

## Discussion

In the state of Amazonas, as well as in other Brazilian states or in other countries, women are less affected by SBEs due to their lower exposure to environments and activities favorable to bites such as fishing, livestock farming, and hunting [3,4]. Although SBEs in women are less

**Table 2. Characteristics of the fatal snakebite envenoming cases in women of childbearing age.**

| Case | Age (years) | Place of residence | Race | Schooling | Perpetrating snake | Time to medical care (in hours) | Site of the bite | Blood clotting time⊃ | Clinical manifestations | Clinical classification on admission, antivenom treatment | Complications | Time until death (in days) | Immediate cause of death§ |
|---|---|---|---|---|---|---|---|---|---|---|---|---|---|
| 1# | 39 | Tabatinga, rural | Amerindian | Unknown | Bothrops | Unknown | Foot | Unknown | Unknown | Severe, 8 BA vials | Unknown | 1 | Unknown |
| 2 | 44 | Japurá, rural | Pardo | Incomplete elementary school | Lachesis | 6 | Foot | Unclottable | Pain, swelling, ecchymosis and vagal syndrome | Severe, 10 BA vials | Necrosis and seconda y bacterial infection | 1 | Sepsis, respiratory distress and cardiac arrest |
| 3 | 37 | Santa Isabel do Rio Negro, rural | Amerindian | Incomplete elementary school | Unknown | Unknown | Leg | Unclottable | Pain and swelling¶ | ... | Unknown | Unassisted death | Unknown |
| 4 | 26 | Coari, rural | Pardo | Incomplete elementary school | Bothrops | 12 | Leg | Unknown | Pain, swelling and respiratory distress | Severe, 5 BA vials | Acute kidney injury | 1 | Respiratory failure |
| 5 | 49 | Manicoré, rural | Amerindian | Unknown | Bothrops | 6 | Leg | Unclottable | Pain, swelling and oliguria | Severe, 8 BA vials | Secondary infection and acute kidney failure | 5 | Acute post-hemorrhagic anemia |
| 6 | 47 | Beruri, rural | Amerindian | Unknown | Bothrops | ... | Finger | Unclottable | Pain, swelling and ecchymosis¶ | ... | Unknown | Unassisted death | Unknow |
| 7 | 45 | Tabatinga, rural | Amerindian | Incomplete elementary school | Bothrops | 3 | Finger | Unknown | Pain and swelling | Mild, 6 BA vials | Unknown | 12 | Stroke |
| 8 | 26 | São Gabriel da Cachoeira, rural | Amerindian | Unknown | Bothrops | > 24 | Leg | Unknown | Pain, swelling, ecchymosis and oliguria | Severe, 6 BA vials | Secondary bacterial infection and acute kidney failure | 3 | Acute respiratory failure, sepsis, acute kidney failure and muscle ischemic infarction |
| 9 | 48 | Manaus, urban | Pardo | Incomplete elementary school | Bothrops | > 24 | Foot | Unknown | Pain and swelling | Severe, 16 BA vials | Secondary bacterial infection | 23 | Sepsis, pneumonia, acute kidney injury |

#Pregnant women, first trimester of pregnancy, unknown number of antenatal consultations.

⊃Lee–White clotting test.

¶Information by interviewing relatives.

§According Mortality Information System (SIM).

Abbreviation–BA: Bothrops antivenom.

**Table 3. Demographic and clinical aspects between pregnancy with complications and no complications.**

| Characteristics | Total, % | No complications n, % | With complications#, n,% | P | OR (95%CI) |
|---|---|---|---|---|---|
| | n = 145 | n = 121 | n = 24 | | |
| Mean age (±SD) | 25 (8.5) | 24.9 (7.9) | 25.7 (10.9) | 0.67 | 1.01 (0.96–1.06) |
| **Ethnicity (n = 145)** | | | | | |
| White | 4 (2.8%) | 3 (2.5%) | 1 (4.2%) | . . . | 1 |
| Black | 5 (3.4%) | 4 (3.3%) | 1 (4.2%) | 0.86 | 0.75 (0.03–17.51) |
| Asian | 1 (0.7%) | 1 (0.8%) | 0 (0.0%) | . . . | 1 |
| *Pardo* | 90 (62.1%) | 76 (62.8%) | 14 (58.3%) | 0.62 | 0.55 (0.05–5.70) |
| Amerindian | 44 (30.3%) | 36 (29.8%) | 8 (33.3%) | 0.74 | 0.67 (0.06–7.27) |
| **Education (in years) (n = 104)** | | | | | |
| 0–4 | 12 (11.5%) | 9 (10.8%) | 3 (14.3%) | . . . | 1 |
| 4–7 | 58 (55.8%) | 44 (53.0%) | 14 (66.7%) | 0.95 | 0.95 (0.22–4.02) |
| 8–15 | 27 (25.9%) | 24 (28.9%) | 3 (14.3%) | 0.27 | 0.37 (0.06–2.21) |
| 16–18 | 6 (5.8%) | 5 (6.0%) | 1 (4.8%) | 0.69 | 0.60 (0.05–7.41) |
| >18 | 1 (1.0%) | 1 (1.2%) | 0 (0.0%) | . . . | 1 |
| **Ocurrence zone (n = 145)** | | | | | |
| Urban | 22 (15.2%) | 18 (14.9%) | 4 (16.7%) | . . . | 1 |
| Rural | 121 (83.4%) | 101 (83.5%) | 20 (83.3%) | 0.85 | 0.89 (0.27–2.91) |
| Periurban | 2 (1.4%) | 2 (1.7%) | 0 (0.0%) | . . . | 1 |
| **Time from bite to medical assistance (in hours) (n = 141)** | | | | | |
| 0–3 | 76 (53.9%) | 63 (53.8%) | 13 (54.2%) | . . . | 1 |
| 3–12 | 39 (27.7%) | 35 (29.9%) | 4 (16.7%) | 0.33 | 0.55 (0.17–1.83) |
| 12–24 | 11 (7.8%) | 7 (6.00%) | 4 (16.7%) | 0.28 | 2.77 (0.70–10.85) |
| >24 | 15 (10.6%) | 12 (10.3%) | 3 (12.5%) | 0.78 | 1.21 (0.29–4.90) |
| **Work-related bite (n = 135)** | 48 (35.6%) | 38 (34.2%) | 10 (41.7%) | 0.49 | 1.37 (0.56–3.38) |
| **Gestational age at the time of the bite (n = 145)** | | | | | |
| Trimester 1 | 11 (7.6%) | 10 (8.3%) | 1/24 (4.2%) | . . . | 1 |
| Trimester 2 | 26 (17.9%) | 25 (20.7%) | 1/24 (4.2%) | 0.51 | 0.4 (0.02–7.03) |
| Trimester 3 | 13 (9.0%) | 9 (7.4%) | 4/24 (16.7%) | 0.21 | 4.4 (0.41–47.5) |
| Unknown | 95 (65.5%) | 77 (63.6%) | 18/24 (75.0%) | 0.42 | 2.3 (0.28–19.45) |
| **Bite site (n = 145)** | | | | | |

(*Continued*)

**Table 3.** (Continued)

| Characteristics | Total, % | No complications n, % | With complications#, n,% | P | OR (95%CI) |
|---|---|---|---|---|---|
| | **n = 145** | **n = 121** | **n = 24** | | |
| Head | 1 (0.7%) | 1 (0.8%) | 0 (0.0%) | . . . | 1 |
| Upper limbs | 14 (9.7%) | 12 (9.9%) | 2 (8.3%) | 0.84 | 0.85 (0.17–4.12) |
| Lower limbs | 129 (89.0%) | 108 (89.3%) | 21 (87.5%) | . . . | 1 |
| Trunk | 1 (0.7%) | 0 (0.0%) | 1 (4.2%) | . . . | 1 |
| **Local manifestations (n = 136)** | | | | | |
| Pain | 134 (98.5%) | 112 (98.2%) | 22 (100.0%) | . . . | 1 |
| Edema | 108 (80.0%) | 90 (79.6%) | 18 (81.8%) | 0.82 | 1.15 (0.36–3.73) |
| Ecchymosis | 20 (15.0%) | 14 (12.6%) | 6 (27.3%) | 0.09 | 2.6 (0.87–7.75) |
| Necrosis | 1 (0.8%) | 1 (0.8%) | 0 (0.0%) | . . . | 1 |
| Secondary bacterial infections | 1 (0.8%) | 1 (0.8%) | 0 (0.0%) | . . . | 1 |
| Acute kidney injury | 2/21 (9.5%) | 2/18 (11.1%) | 0 (0.0%) | . . . | 1 |
| **Unclottable blood (n = 88)¶** | 37 (42.0%) | 28 (38.9%) | 9 (56.3%) | 0.21 | 2.02 (0.68–6.04) |
| **Snake genus (n = 145)** | | | | | |
| *Bothrops* | 109 (75.2%) | 92 (76.0%) | 17 (70.8%) | . . . | 1 |
| *Lachesis* | 22 (15.2%) | 17 (14.0%) | 5 (20.8%) | 0.42 | 1.59 (0.52–4.90) |
| Unknown | 14 (9.6%) | 12 (9.9%) | 2/24 (8.4%) | 0.79 | 1.35 (0.14–12.86) |
| **Severity classification (n = 140)** | | | | | |
| Mild | 60 (42.9%) | 51 (43.6%) | 9 (39.1%) | . . . | 1 |
| Moderate | 71 (50.7%) | 58 (49.6%) | 13 (56.5%) | 0.61 | 1.27 (0.5–3.22) |
| Severe | 9 (6.4%) | 8 (6.8%) | 1 (4.3%) | 0.76 | 0.71 (0.08–6.37) |
| **Antivenom treatment (n = 141)** | 123 (87.2%) | 102 (87.2%) | 21/24 (87.5%) | 0.44 | 0.76 (0.38–1.51) |

#Low birth weight, preterm birth, fetal deaths, and neonatal deaths.

¶Lee-White clotting test.

Abbreviation: SD = standard deviation.

frequent, the consequence for perinatal outcomes should not be neglected. As with non-pregnant women, pregnant women who are bitten by snakes may suffer tissue damage, pain, swelling, tissue necrosis, and functional loss of limbs. More severe effects include hypotension, coagulopathies, falling fibrinogen and platelet levels, as well as pregnancy-related adverse outcomes such as miscarriages, placental abruption, preterm labor, and fetal malformations [20]. Although SBEs in pregnant women have important consequences for both the mother and the fetus or neonate, studies related to the subject are scarce and there are no accurate estimates of the number of cases of SBEs in women of childbearing age and pregnant women. In the

**Table 4. Perinatal outcomes in pregnant women affected by snakebite envenomation and other pregnant women.**

| Variables | Overall | Snakebite envenomation during pregnancy | | Univariate | | Multivariate# | |
|---|---|---|---|---|---|---|---|
| | | No | Yes | P | OR (95%CI) | P | AOR (95% CI) |
| Low birth weight | 87,276/1,228,463 (7.1%) | 87,268/1,228,328 (7.1%) | 8/135 (5.9%) | 0.48 | 0.78 (0.35–1.48) | 1 | . . . |
| Preterm birth | 118,879/1,209,939 (9,8%) | 118,865/1,209,801 (9.8%) | 14/138 (10.0%) | 0.90 | 1.04 (0.57–1.74) | 1 | . . . |
| Perinatal death | 26,187/1,263,650 (2.1%) | 26,179/1,263,505 (2.1%) | 8/145 (5.6%) | <0.01 | 2.76 (1.24–5.27) | <0.01 | 2.58 (2.21–2.99) |
| Fetal death | 13,569/1,263,650 (1.1%) | 13,565/1,263,505 (1.1%) | 4/145 (2.8%) | 0.06 | 2.61 (0.80–6.19) | <0.01 | 2.24 (1.80–2.76) |
| Neonatal death | 12,618/1,263,650 (1.0%) | 12,614/1,263,505 (1.0%) | 4/145 (2.8%) | 0.04 | 2.81 (0.86–6.66) | <0.01 | 2.89 (2.34–3.52) |

#Adjusted for age and education, and number of antenatal consultations.

present study, we estimated the number of women of childbearing age and pregnant women who suffered SBEs in the state of Amazonas over a period of 15 years, using data reported on SINAN. In addition, this study highlighted the clinical-demographic descriptions and risk factors for complications of SBEs during pregnancy.

During the study period, 3,297 SBEs occurred in women of childbearing age and 274 (8.3%) in pregnant women. Case-fatality rates were similar between pregnant and non-pregnant women, and fetal or neonatal losses were low, but present. Two hundred thirteen SBEs in pregnant women were reported in the literature between 1966 and 2009, with an overall case-fatality rate of 4%, and a fetal loss rate of 20% [20]. Unlike the rates reported in the literature, it is important to consider that the real number of SBEs can be underestimated, although notification has been mandatory in Brazil (2). Failure to seek medical assistance due to the absence of symptoms, distance to the health care unit, and the use of traditional medicinal practices may be related to the low notification of cases [13,14].

In pregnant women, delay in proper clinical management can result in poor outcomes, such as death of both the mother and the fetus or neonate, or morphological alterations, such as hydrocephalus and deformations [20,28]. We did not find differences in the time taken to seeking care between pregnant and non-pregnant women. However, it is important to highlight that woman tended to seek medical assistance in the first 3 hours after the accident, a time that is considered short when compared to SBEs in men in the same region [3,4]. For pregnant women, access to care and use of antivenom is time-dependent [20]. There is no restriction for antivenom administration in pregnant women [20]. In this study, however, twenty-five percent of pregnant women did not receive antivenom, which is comparable to non-pregnant women.

In the Amazon, the vast majority of SBEs are caused by *Bothrops atrox* [9]. *Bothrops* envenomations induce important physiological imbalances in coagulation, blood pressure, and renal and respiratory functions [15,29]. Respiratory failure, obstetric bleeding with progression to anemia, septicemia, secondary infection, and acute renal failure are the main complications in pregnant women who have suffered SBEs [20,30–32]. Although these complications were found in pregnant women, they did not differentiate between pregnant and non-pregnant, potentially due to a balance between factors associated with envenomations and those associated with pregnancy. During pregnancy, the pregnant woman undergoes significant anatomical and physiological changes, which include (i) an increase in plasma volume with a

**Table 5. Characteristics of the perinatal deaths related to snakebite envenomation in pregnant women.**

| Case | Fetal or neonatal information | | | | | | Maternal information | | | | | | | | |
|---|---|---|---|---|---|---|---|---|---|---|---|---|---|---|---|
| | Place of birth | Gender | Lifetime | Weight (grams) | Apgar# | Pregnancy time (week) | Cause of death | Age | Place of residence | Ethnicity | Schooling | Number of antenatal consultations, type of delivery | Perpetrating snake | Clinical classification on admission, antivenom treatment | Site of the bite | Clinical description and blood clotting time◌ | Gestational trimester of the snakebite |
| **Fetal deaths** | | | | | | | | | | | | | | | | | |
| 1 | Residence | M | … | ND | … | ≥42 | Labor complications and congenital malformations | 46 | Maués, rural | Pardo | Complete elementary school | 0, vaginal | Bothrops | Mild, 5 BLA vials | Foot | Unknown clinical signs, normal blood clotting | 1 |
| 2 | Hospital | F | … | 1,850 | … | 32–36 | Unspecified cause | 18 | Urucurituba, urban | Pardo | Complete elementary school | 0, vaginal | Bothrops | Mild, 10 BA vials | Foot | Pain and swelling, and unclottable blood | 1 |
| 3 | Hospital | F | … | 766 | … | 22–27 | Placental detachment and hemorrhage, and maternal trauma | 15 | Parintins, rural | Pardo | Complete elementary school | 0, vaginal | Unknown | Mild, no | Foot | Pain and swelling | 1 |
| 4 | Hospital | M | … | 830 | … | 22–27 | Extreme immaturity | 40 | Jutaí, urban | Pardo | Complete elementary school | 0, vaginal | Bothrops | Mild, 4 BA vials | Leg | Pain, swelling, ecchymosis, hypotension, vomiting, and normal blood clotting | 1 |
| **Neonatal deaths** | | | | | | | | | | | | | | | | | |
| 5 | Hospital | M | 5 h | 3,200 | ND | 37–41 | Bacterial sepsis and respiratory failure | 42 | Borba, rural | Amerindian | Complete elementary school | 0, vaginal | Bothrops | Mild, 5 BLA vials | Foot | Pain, swelling, and unclottable blood | 1 |
| 6 | Hospital | M | 4 h | 1,135 | 4;7 | 22–27 | Low weight | 16 | Parintins, rural | Pardo | Complete elementary school | 0, vaginal | Lachesis | Moderate, 8 BLA vials | Foot | Pain, swelling, ecchymosis, and unclottable blood | 1 |
| 7 | Hospital | F | 16 h | 2,150 | 10;8 | 32–36 | Heart malformation, congenital renal failure, and bacterial septicemia | 39 | Pauini, rural | Black | Illiterate | 1, Cesarean | Bothrops | Moderate, 8 BA vials | Leg | Pain and swelling | 2 |
| 8 | Hospital | M | 1 h | 315 | ND | <22 | Respiratory failure, neonatal meconium aspiration, and bacterial septicemia | 14 | São Gabriel da Cachoeira, rural | Amerindian | Unknown | 0, Cesarean | Bothrops | Severe, 12 BLA vials | Forearm | Pain, swelling and ecchymosis | 1 |

F: female, M: male; h: hours; ND: not defined

#At 1st and 5th minutes, respectively.

§According to the Mortality Information System (SIM).

◌Lee-White clotting test.

Abbreviation–BA: Bothrops antivenom; BLA: Bothrops-Lachesis antivenom.

consequent increase in the mass of red blood cells; and a drop in the concentration of hemoglobin, hematocrit and red blood cell counts; (ii) alterations of the coagulation with a physiological state of hypercoagulability; (iii) adaptive changes in the renal vasculature; (iv) increased renal plasma flow and glomerular filtration rate and others [33]. These physiologic changes may have helped counterbalance the pathologic changes due to SBEs, leading to the absence of differential clinical symptoms between pregnant and non-pregnant women.

Although the major function of the placenta is to transfer nutrients and oxygen from the mother to the fetus and to assist in the removal of waste products from the fetus to the mother, it is now known that the placenta is not impenetrable against drugs and toxic substances [34]. In theory, only small molecules (<600 Daltons) can easily transit human placenta [35]. A case series study showed that snake venom crosses the placenta and in amounts that, though insufficient to cause systemic envenoming in the mother, can cause systemic envenoming in the fetus [36], which could be caused by some suggested mechanisms (e.g., simple diffusion, facilitating diffusion, active transport, or pinocytosis) [37]. The greatest risk of adverse toxin-induced effects on the fetus is probably during organogenesis, which takes place in the first trimester. Thus, pregnant women who suffered SBEs were 2.19 and 2.79 times more at risk for fetal death and neonatal death, respectively. On the other hand, not only direct effects of the venom may be responsible for fetus outcomes, fetal hypoxia due to maternal shock, placental bleeding due to coagulopathy, venom-induced uterine contractions, and pyrexia and cytokine release are likely to also involved [38].

Additionally, although heterologous IgGs are known to be transported through FcRn receptors (i.e., could cross the placenta and reach the fetus), it is known that Fab and F(ab')2 antivenoms cannot cross the placenta [39,40] and, in Brazil, antivenoms are all composed of F(ab')2 heterologous antibodies [41]. In order to overcome the hypersensitivity reactions caused by the horse-derived antibodies, many antivenom manufacturers (including Butantan Institute) digest the whole IgG using pepsin to produce bivalent F(ab')2 fragments, which lack the C terminal Fc domains. Indeed, the digestion is reported to reduce complex-mediated or type III hypersensitivity reactions; however, the molecule retains divalency and the ability to bind complement. It is neither possible to predict whether the pregnancy outcomes were related to the direct effect of the venom on the fetus/placenta, nor if it is an indirect effect caused by immune system activation and inflammation, but we assume that both may participate in the process.

The causes attributed to death were septicemia, respiratory failure and/or distress, specified complications of labor, birth defects, fetus and neonates affected by maternal trauma, placental abruption and hemorrhage, renal failure, and meconium aspiration. The fetus of pregnant women who have suffered SBEs are at severe risk of hypoxia, and death [42,43]. Complications from fetal hypoxia/anoxia are among the leading causes of fetal death [44]. Hypoxia, secondary to placental dysfunction, plays an important role in most fetal deaths, but the evidence is indirect and an understanding of the causes of hypoxia and the intermediate steps between fetal hypoxia and fetal death are necessary [45]. The consequence of hypoxia is the failure of the fetus to achieve its genetically determined growth potential. Intrauterine growth restriction is associated with distress and asphyxia and a 6- to 10-fold increase in perinatal death [46–47]. Despite the worldwide frequency of stillbirths due to different conditions, such as infectious diseases, the subsequent implications are neglected and underestimated [48]. Fetal or neonatal losses are painful experiences that have biological, psychological, social, and spiritual consequences for parents and family members [49,50]. The indirect and intangible costs of stillbirths are extensive and are usually covered only by families. This issue is particularly onerous for those with little resources [50], such rural populations in the interior of the Amazon.

This study had major limitations related to the completeness of the information of pregnancy outcomes. Nearly half of pregnant women with reports of SBEs during pregnancy had no outcome information available in the databases. The possible reasons for this difference are the limitations of the surveillance of SBE cases due to errors in self-reporting pregnancy during SBE reporting, or due to fetal deaths prior to week 20 of pregnancy, which are not reported in *SIM* or *SINASC*. In Brazil, underreporting of live birth records in *SINASC* has already been verified in the Extra-Amazonian Region of the country, especially in municipalities with less health infrastructure [51]. Although we do not have estimates of underreporting of live birth records for the Brazilian Amazon, we believe that these should be higher than in the rest of the national territory since, among traditional populations, especially among indigenous peoples, which are heavily affected by SBEs, deliveries occurring at home are very common. Attempts to contact the pregnant women failed due to the lack of phone contacts or addresses in the databases. Another way of accessing this information would be from medical records. However, as we highlighted above, most of them must have given birth at home, and do not have hospital records. Moreover, the storage of hospital information in remote towns of the Amazon is very precarious, which is a bottleneck for the collection of medical information. However, the pregnant women with outcome information and those with missing data on outcome were similar in relation to sociodemographic variables, suggesting that missing data may not be so biased.

## Conclusion

Although the incidence of SBEs in pregnant women is low, the associated risk of adverse outcomes such as maternal death, fetal demise and neonatal death are increased and cannot be neglected.

## Supporting information

**S1 File. STROBE Checklist for Observational Studies.**
(DOC)

**S2 File. Study database.**
(XLSX)

## Acknowledgments

We would like to the technicians of the *Fundação de Vigilância em Saúde do Amazonas Dra. Rosemary Costa Pinto* for providing the information used in this study.

## Author Contributions

**Conceptualization:** Djane Clarys Baia-da-Silva, Jacqueline Sachett, Marco A. Sartim, Fan Hui Wen, Manuela B. Pucca, Charles J. Gerardo, Vanderson S. Sampaio, Priscila Ferreira de Aquino, Wuelton M. Monteiro.

**Data curation:** Alexandre Vilhena Silva-Neto, Patrícia Carvalho da Silva Balieiro, Lisele Brasileiro.

**Formal analysis:** Alexandre Vilhena Silva-Neto, Patrícia Carvalho da Silva Balieiro, Antônio Alcirley da Silva Baleiro, Lisele Brasileiro, Vanderson S. Sampaio, Wuelton M. Monteiro.

**Investigation:** Thaís P. Nascimento, Djane Clarys Baia-da-Silva, Antônio Alcirley da Silva Baleiro, Wuelton M. Monteiro.

**Methodology:** Alexandre Vilhena Silva-Neto, Patrícia Carvalho da Silva Balieiro, Flor Ernestina Martinez-Espinosa, Priscila Ferreira de Aquino, Wuelton M. Monteiro.

**Project administration:** Thaís P. Nascimento, Priscila Ferreira de Aquino, Wuelton M. Monteiro.

**Resources:** Vanderson S. Sampaio, Priscila Ferreira de Aquino, Wuelton M. Monteiro.

**Software:** Alexandre Vilhena Silva-Neto.

**Supervision:** Antônio Alcirley da Silva Baleiro, Jacqueline Sachett, Flor Ernestina Martinez-Espinosa, Vanderson S. Sampaio, Priscila Ferreira de Aquino, Wuelton M. Monteiro.

**Validation:** Patrícia Carvalho da Silva Balieiro, Antônio Alcirley da Silva Baleiro, Wuelton M. Monteiro.

**Visualization:** Alexandre Vilhena Silva-Neto, Wuelton M. Monteiro.

**Writing – original draft:** Thaís P. Nascimento, Djane Clarys Baia-da-Silva, Marco A. Sartim, Wuelton M. Monteiro.

**Writing – review & editing:** Jacqueline Sachett, Lisele Brasileiro, Marco A. Sartim, Flor Ernestina Martinez-Espinosa, Fan Hui Wen, Manuela B. Pucca, Charles J. Gerardo, Vanderson S. Sampaio, Priscila Ferreira de Aquino.

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
