## [Decision Letter · Decision Letter 0]

7 Oct 2022

Dear Dr Monteiro,

Thank you very much for submitting your manuscript "Pregnancy Outcomes After Snakebite Envenomations: A Retrospective Cohort in the Brazilian Amazonia" for consideration at PLOS Neglected Tropical Diseases. As with all papers reviewed by the journal, your manuscript was reviewed by members of the editorial board and by several independent reviewers. In light of the reviews (below this email), we would like to invite the resubmission of a significantly-revised version that takes into account the reviewers' comments. 

We cannot make any decision about publication until we have seen the revised manuscript and your response to the reviewers' comments. Your revised manuscript is also likely to be sent to reviewers for further evaluation.

Sincerely,

Abdulrazaq G. Habib

Guest Editor

José María Gutiérrez

Section Editor

Reviewer's Responses to Questions

**Key Review Criteria Required for Acceptance?**

**Methods**

-Are the objectives of the study clearly articulated with a clear testable hypothesis stated?

-Is the study design appropriate to address the stated objectives?

-Is the population clearly described and appropriate for the hypothesis being tested?

-Is the sample size sufficient to ensure adequate power to address the hypothesis being tested?

-Were correct statistical analysis used to support conclusions?

-Are there concerns about ethical or regulatory requirements being met?

Reviewer #1: Clear objectives, Clear study design. 

But info missing on half of the pregnant patients with SBE, see general comments below.

Reviewer #2: Introduction 

I am unfamiliar with the meaning of the term ‘anticipation of labor.’ Is this referring to preterm labor? Suggest changing this to more established terminology.

Methods

Change ‘one thousand nine-hundred and ninety-six’ to 1,996.

The definition of perinatal mortality differs from the WHO definition which is cited in reference 26. I believe the established definition of perinatal mortality is death from 22-weeks gestation until 7 days after birth. I would suggest using a different term (such as ‘combined fetal and neonatal mortality’) or changing the analysis to align with the accepted definition of perinatal mortality.

Reviewer #3: The study is on an important aspect of snakebite, on which there is little information. The objectives are clear and the study population clearly defined. The sample size is adequate, and I have no ethical concerns. 

However, studies with secondary and retrospective databases have many limitations regarding completeness of information, especially those from remote rural areas in LMIC. This study is no different. Nearly half of the pregnant women with reports of snakebite envenoming during pregnancy had no pregnancy outcome information – this is a major limitation.

**Results**

-Does the analysis presented match the analysis plan?

-Are the results clearly and completely presented?

-Are the figures (Tables, Images) of sufficient quality for clarity?

Reviewer #1: See general comments section

Reviewer #2: My main concern is the large proportion of missing data on pregnancy outcome (around 50%). Pregnancy outcome is the headline finding of the paper and this amount of missing data could have a substantial impact on the result. It seems this was dealt with by listwise deletion and there is no justification as to why this approach was used. Was the data 'missing at random?' If not missing at random, perhaps multiple imputation would have been a better way of dealing with this?

Other minor comments:

Change ‘Three thousand two-hundred and ninety-seven’ to 3,297.

There is no mention of increased risk of skin necrosis amongst pregnant women in the results section, but this is in the abstract. In table 1 there seems to be no increased risk of skin necrosis between pregnant and non-pregnant women.

Can population incidence of snakebite in pregnancy be calculated? This depends on whether denominator would accurately reflect the total number of pregnancies in the study area during the study period. If this wouldn’t be accurate, then ignore this comment.

Table 5 – could the number of days between the snakebite and the perinatal death be added?

Reviewer #3: The analysis is appropriate and the data is clearly presented. The Tables are of sufficient clarity.

**Conclusions**

-Are the conclusions supported by the data presented?

-Are the limitations of analysis clearly described?

-Do the authors discuss how these data can be helpful to advance our understanding of the topic under study?

-Is public health relevance addressed?

Reviewer #1: see general comments section

Reviewer #2: It is suggested that placental transfer of venom toxins to the fetus is the most likely explanataion for the higher fetal mortality rate that was observed. The reference (36) for venoms crossing the placenta is a case report of a pregnant woman with snakebite complicated by DIC and placental abruption which resulted in still birth – I cannot see that this demonstrates fetal transfer of toxins (this was fetal death due to abruption). Reference 37 is a general review on drug transfer across the placenta – but not venom toxins. Although evidence is lacking, on balance it would seem unlikely that the large venom toxin proteins would be able to cross the placenta. Damage to the placenta (on the maternal side of the ciculation) seems a more likely mechanism leading to poor fetal outcome. 

Reference 41 does not seem to be relevant to the statement on fetal coagulopathy (in the case report cited the infant had multiple abnormalities that could have various underlying causes, including microcephaly, hyperbilirubinaemia, anaemia, and thrombocytopaenia. I believe the newborn did not have a coagulopathy).

Large proportion of missing data on pregnancy outcome should be highlighted in the discussion with a mention of how this may have affected the results. This could have had a substantial impact on the reported rate of fetal/neonatal death.

Reviewer #3: It may be difficult to arrive at firm conclusions with a significant limitation of information regarding pregnancy outcomes in snakebite victims (nearly half). The limitations should be discussed in more detail and not limited to a short paragraph at the end of the discussion. 

The Discussion section is otherwise well written and addresses the public health importance of the topic. However, there has been only a feeble attempt to compare, contrast, and discuss work in the area from other parts of the world.

**Editorial and Data Presentation Modifications?**

Reviewer #1: see general comments

Reviewer #2: (No Response)

Reviewer #3: There are several minor typographical errors, but overall the paper is well written, and I enjoyed reading it.

**Summary and General Comments**

Reviewer #1: The manuscript discusses pregnancy & snakebite envenoming which is highly needed data. There are nevertheless two major challenges I see based on the manuscript. For half of the pregnant women with a snakebite envenoming there were no pregnancy outcomes available. The authors repot one of the reasons of unknown outcomes is fetal deaths before week 20 which are not included in the used databases. That suggests that there may be a selection included in the manuscript not reflecting the overall outcomes. The residence of the victims is known; is it with further ethical approval possible to obtain the hospital data from these missing patients? Even if the outcomes of half of these missings can be obtained, that would be useful.

Authors also describe the antivenom currently used is not crossing the placenta. But the study described a 15 years period; do the authors know what is used earlier? And if not, how do we than know if the pregnancy outcomes are related to the snakebite envenoming instead of the antivenom/allergic reactions? 

Further suggestions are below. 

Intro:

Line 96: You describe the clinical presentation of Bothrops bites but there is no info how common Bothrops bites are in Brazil. This follows in line 151. Suggest to combine that background. It deserves a place in the introduction and not only in the methods. 

Line 99-100: can you check this sentence? It’s a bit hard to read. 

Line 133: Can you provide more info on the databases used? How many persons are in the database? And what about privacy? And data management? Who did the blocking? In how many pregnant women were you not able to find a match apart from the pregnancy outcomes? 

Line 158-170: variables collected but not clear yet how these relate to pregnancy and SBE, eg why is data on race/ethnicity collected? 

The data are from 2007-2021; any changes in that period when thinking about pregnancy outcomes and the Zika time? Apart from zika, how does are the negative outcomes placed in time? Are they more often > 10 years ago or distributed over time evenly? 

Line 207: why does gender need to be included here?

Line 246: instead of a p-value, a difference in age with CI would be more informative here. 

Line 253: How was the diagnosis Bothrops made? And can you add info in the introduction on the number of dry bites in Bothrops bites? 

Line 254-255: how do these percentages relate to percentages in child bearing age/not pregnant? How is unclottable blood defined? 

Line 263: remove p value, keep OR and CI, same for OR’s in rest of manuscript. Once we have the OR and CI the pvalue is of no added benefit. 

Line 266: since numbers are so low, include absolute values apart from percentages. 

Line 287: remind us how many have missing outcomes and include in discussion (see first lines of review)

293: deaths before week 20 would be crucial info to get.

302/303: how many of the patients with poor outcomes did receive antivenom? Include in results in summarized way. 

307: talks about 3 deaths, but in line 309 and 312 talks about four cases. 

Line 350: provide CI of the 4%. 

Tables: why is the third table needed? In the first two tables I suggest to replace all variables presented in categorized way by the data as collected. Age was not collected in categories, so you can calculate a median with IQR. We don’t need the categories. Same for some other variables and it provides more data than the categories. I noticed there was Y where it should be AND.

Reviewer #2: Overall I thought this was an excellent study and provides a rare insight into pregnancy outcome in snakebite. The large samples size and representative population are strengths. The main limitation is the large proportion of missing on pregnancy outcomes. I feel there needs to be more careful consideration of how best to deal with this missing data.

Reviewer #3: The paper discusses a neglected area of snakebite, where there are misconceptions on the management even among physicians (eg. use of antivenom). It is clearly and important topic. The major limitation is use of secondary data which the authors themselves admit may be incomplete and unreliable in the study setting. The results clearly show this, with a significant gap in data on pregnancy outcome - an important objective of the study. The authors should to describe work in this area from other parts of the world, even though they are small studies.

PLOS authors have the option to publish the peer review history of their article (what does this mean?). If published, this will include your full peer review and any attached files.

Reviewer #1: No

Reviewer #2: Yes: Michael Abouyannis

Reviewer #3: No
---

## [Decision Letter · Decision Letter 1]

13 Nov 2022

Dear Dr Monteiro,

Thank you very much for submitting your manuscript "Pregnancy Outcomes after Snakebite Envenomations: A Retrospective Cohort in the Brazilian Amazonia" for consideration at PLOS Neglected Tropical Diseases. As with all papers reviewed by the journal, your manuscript was reviewed by members of the editorial board and by several independent reviewers. The reviewers appreciated the attention to an important topic. Based on the reviews, we are likely to accept this manuscript for publication, providing that you modify the manuscript according to the review recommendations. 

Sincerely,

Abdulrazaq G. Habib

Guest Editor

José María Gutiérrez

Section Editor

Reviewer's Responses to Questions

**Key Review Criteria Required for Acceptance?**

**Methods**

-Are the objectives of the study clearly articulated with a clear testable hypothesis stated?

-Is the study design appropriate to address the stated objectives?

-Is the population clearly described and appropriate for the hypothesis being tested?

-Is the sample size sufficient to ensure adequate power to address the hypothesis being tested?

-Were correct statistical analysis used to support conclusions?

-Are there concerns about ethical or regulatory requirements being met?

Reviewer #1: As a reviewer I am not convinced by the data though I understand it's been a lot of work to get this data, the authors do not explain why it is not possible to try and get the other 50% of the data on outcomes. They do explain that the characteristics of the pregnancies in SBE victims were similar to the other 50% of the data, but that is not my main worry. 

There were 4 neonatal deaths in the current group. Only 4 deaths to base the Odds ratios on. My problem is that such a low number of deaths may be influenced both by chance or selection (not based on characteristics of the mother). Sick mothers may present to the hospital -> included in the numbers, whereas the home births may actually have lower deaths. Or even by chance, the number of deaths in the other half of the population under study may have been between 0 and 6 which would have influenced OR heavily. 

Furthermore, pregnancy outcomes < 20 weeks are not available. Whereas almost all women with fetal death or neonatal death were bitten in the first trimester... How can the authors explain that they link the neonatal death with a bacterial sepsis after birth to the snakebite in the first trimester. It is biologically more difficult to explain than having a -by chance finding- or a selection bias here; the number of deaths higher in the counted SBE victims than in the non-counted SBE victims (who did not present to the hospital). 

The abstract does not well present the challenges with the data.

Reviewer #3: (No Response)

**Results**

-Does the analysis presented match the analysis plan?

-Are the results clearly and completely presented?

-Are the figures (Tables, Images) of sufficient quality for clarity?

Reviewer #1: (No Response)

Reviewer #3: (No Response)

**Conclusions**

-Are the conclusions supported by the data presented?

-Are the limitations of analysis clearly described?

-Do the authors discuss how these data can be helpful to advance our understanding of the topic under study?

-Is public health relevance addressed?

Reviewer #1: (No Response)

Reviewer #3: In the revised version of the paper, the authors have expanded the paragraph on limitations of the study. However, given that losing 50% of data is a major problem as commented on by all three reviewers, they must emphasize this limitation more. They should state that this is a MAJOR limitation, and state that it may be difficult to arrive at firm conclusions with lack of information regarding pregnancy outcomes in nearly half the snakebite victims

**Editorial and Data Presentation Modifications?**

Reviewer #1: (No Response)

Reviewer #3: (No Response)

**Summary and General Comments**

Reviewer #1: (No Response)

Reviewer #3: The authors must emphasize the limitation of losing information on nearly half the pregnant women more. They should state that this is a MAJOR limitation. See comment in section on Conclusions.

PLOS authors have the option to publish the peer review history of their article (what does this mean?). If published, this will include your full peer review and any attached files.

Reviewer #1: No

Reviewer #3: Yes: Prof. H. Janaka de Silva

Figure Files:

Data Requirements:

Reproducibility:

References

---

## [Editor Report · Decision Letter 2]

18 Nov 2022

Dear Dr Monteiro,

We are pleased to inform you that your manuscript 'Pregnancy Outcomes after Snakebite Envenomations: A Retrospective Cohort in the Brazilian Amazonia' has been provisionally accepted for publication in PLOS Neglected Tropical Diseases.

Best regards,

Abdulrazaq G. Habib

Guest Editor

José María Gutiérrez

Section Editor

---

## [Editor Report · Acceptance letter]

29 Nov 2022

Dear Dr. Monteiro,

We are delighted to inform you that your manuscript, "Pregnancy Outcomes after Snakebite Envenomations: A Retrospective Cohort in the Brazilian Amazonia," has been formally accepted for publication in PLOS Neglected Tropical Diseases.

Best regards,

Shaden Kamhawi

co-Editor-in-Chief

Paul Brindley

co-Editor-in-Chief
